# Microenvironmental Factors in Oral Cavity Squamous Cell Carcinoma Undergoing Surgery: Correlation with Diffusion Kurtosis Imaging and Dynamic Contrast-Enhanced MRI

**DOI:** 10.3390/cancers15010015

**Published:** 2022-12-20

**Authors:** Antonello Vidiri, Andrea Ascione, Francesca Piludu, Eleonora Polito, Enzo Gallo, Renato Covello, Paola Nisticò, Vittoria Balzano, Barbara Pichi, Raul Pellini, Simona Marzi

**Affiliations:** 1Radiology and Diagnostic Imaging Department, IRCCS Regina Elena National Cancer Institute, Via Elio Chianesi 53, 00144 Rome, Italy; 2Department of Radiological, Oncological and Pathological Sciences, Sapienza, University of Rome, Viale Regina Elena 324, 00161 Rome, Italy; 3Scuola di Specializzazione in Radiodiagnostica, Sapienza Università di Roma—Policlinico Umberto I, Viale Regina Elena 324, 00161 Rome, Italy; 4Pathology Department, IRCCS Regina Elena National Cancer Institute, Via Elio Chianesi 53, 00144 Rome, Italy; 5Tumor Immunology and Immunotherapy Unit, IRCCS Regina Elena National Cancer Institute, Via Elio Chianesi 53, 00144 Rome, Italy; 6Otolaryngology & Head and Neck Surgery, Regina Elena National Cancer Institute, Via Elio Chianesi 53, 00144 Rome, Italy; 7Medical Physics Laboratory, IRCCS Regina Elena National Cancer Institute, Via Elio Chianesi 53, 00144 Rome, Italy

**Keywords:** oral cavity cancer, squamous cell carcinoma, diffusion magnetic resonance imaging, perfusion magnetic resonance imaging, tumor microenvironment

## Abstract

**Simple Summary:**

Although the potential role of imaging-based parameters or pathological factors as useful cancer biomarkers has been reported in several studies, only a few investigations have been conducted to assess the associations between them. We hypothesized that the imaging features of OSCC, derived from perfusion- and diffusion-weighted techniques, may reflect not only the characteristics of cancer but also the microenvironment composition. We found that a number of pathological factors affected the imaging parameters, i.e., the inflammatory infiltrate, tumor grading and desmoplastic reaction. Evident relationships emerged between the DKI parameters and tumor-infiltrating lymphocytes. Significant differences in a few DCE-MRI parameters were also found according to the tumor grading. These findings may be helpful for a more comprehensive biological characterization of OSCC and could be used in future investigations for diagnostic and prognostic purposes.

**Abstract:**

Background: In this prospective study, we hypothesized that magnetic resonance imaging (MRI) may represent not only the tumor but also the microenvironment, reflecting the heterogeneity and microstructural complexity of neoplasms. We investigated the correlation between both diffusion kurtosis imaging (DKI) and dynamic contrast-enhanced (DCE)-MRI with the pathological factors in oral cavity squamous cell carcinomas (OSCCs). Methods: A total of 37 patients with newly diagnosed OSCCs underwent an MR examination on a 3T system. The diffusion coefficient (D), the kurtosis parameter (K), the transfer constants K^trans^ and K_ep_ and the volume of extravascular extracellular space v_e_ were quantified. A histogram-based approach was proposed to investigate the associations between the imaging and the pathological factors based on the histology and immunochemistry. Results: Significant differences in the DCE-MRI and DKI parameters were found in relation to the inflammatory infiltrate, tumor grading, keratinization and desmoplastic reaction. Relevant relationships emerged between tumor-infiltrating lymphocytes (TILs) and DKI, with lower D and higher K values being associated with increased TILs. Conclusion: Although a further investigation is needed, these findings provide a more comprehensive biological characterization of OSCCs and may contribute to a better understanding of DKI-derived parameters, whose biophysical meaning is still not well-defined.

## 1. Introduction

Oral cavity squamous cell carcinomas (OSCCs) are the most common malignancy of the oral cavity. Many risk factors such as tobacco products, alcohol, diet and a genetic predisposition are involved in the tumorigenesis of head and neck squamous cell carcinomas (HNSCCs), individually or together, counting for their increased incidence in both developing and developed countries [1].

Magnetic resonance imaging (MRI) and computed tomography (CT) are the preferred imaging techniques for the staging, follow-up and response treatment evaluation of oral cancers, either after surgery or radio-chemotherapy, thanks to the high soft tissue contrast of MRI and the superior ability of CT to detect mandible infiltrations and cortical involvements [2,3]. In addition to conventional images, functional imaging techniques such as diffusion-weighted imaging (DWI) and dynamic contrast-enhanced (DCE)-MRI can give useful information concerning both the tumor and its microenvironment, allowing a global assessment of the neoplasm. The imaging-derived information can be regarded as complementary to the pathological data provided by biopsy sampling, which, in a few cases, can misrepresent the complexity and heterogeneity of malignancies [4,5].

DWI is a quantitative technique that investigates the tissue microstructure through the measurement of the random water molecule diffusion within the tissues by the apparent diffusion coefficient (ADC). However, several variables may influence the ADC, i.e., the most common cellular density, the integrity of cellular membranes, the entity of interstitial space, the tissue perfusion and the tortuosity of the vessels, making the biophysical interpretation of the ADC rather complex [6].

Diffusion kurtosis imaging (DKI) is an advanced DWI technique that focuses on the slow displacement of the water molecules observed at very high *b*-values (>1000 s/mm^2^), resulting in a non-Gaussian diffusion behavior that is supposed to be strictly related to the heterogeneity of the tissue microstructures [7]. Two new parameters have been introduced by the fitting of the DKI signal at increasing b-values: the diffusion coefficient corrected for the non-Gaussian diffusion behavior (D) and the diffusional kurtosis parameter (K), whose biophysical meaning is still not well-defined [8,9].

The hemodynamic characterization of tumors is available via perfusion imaging techniques such as DCE-MRI, which analyzes the blood volume, blood flow and vascular permeability, providing quantitative and/or semi-quantitative perfusion parameters. These perfusion-related parameters have been demonstrated to have great potential in the differential diagnosis, early monitoring and response assessment to chemo-radiotherapy in head and neck cancers [5,10,11].

In addition to cancer cells, solid tumors are made up of various non-malignant cell types and an extracellular matrix, collectively called the tumor microenvironment (TME), which plays a key role in the cancer progression and treatment response. The tumor and its TME are in constant communication and their relationship allows tumor growth and proliferation, angiogenesis, immune escape and metastasis [1]. The study of the TME has become a field of great interest in recent years because of its role in the tumor evolution. A deeper knowledge of the crosstalk between the TME and neoplastic cells is fundamental for a better understanding of the pathological basis of carcinogenesis, for the detection of reliable prognostic markers and for the development of novel immunotherapeutic approaches.

The cellular constituent of the TME is mainly composed of immune cells from the adaptive (lymphocytes) and innate (i.e., macrophages, neutrophils and dendritic cells) systems, together with cancer-associated fibroblasts and endothelial cells. Lymphocytes are known to be a crucial component of the TME; they can generate, together with stromal cells, an immunological tolerance that protects cancer cells and modulates the efficacy of therapies. Immunological biomarkers, in conjunction with other factors related to the microenvironment such as hypoxia, have been recognized to significantly affect the behavior of tumors in response to chemo-radiotherapy and to have a prognostic and predictive value in HNSCCs [12].

Although the potential role of imaging-based parameters or pathological factors as useful cancer biomarkers has been reported in several studies, only a few investigations have been conducted to assess the associations between them [6,13,14,15,16]. Specifically, no clear evidence has emerged in the literature concerning the possible role of diffusion and perfusion MRI in depicting the TME. The purpose of the present study was to assess the relationships between the parameters obtained from both DKI and DCE-MRI with the pathological factors of OSCCs by histology and immunochemistry.

## 2. Materials and Methods

### 2.1. Patient Population

From July 2018 to February 2022, a total of 37 patients admitted to the Department of Radiology and Diagnostic Imaging of IRCCS Regina Elena National Cancer Institute were consecutively enrolled in our prospective study. The inclusion criteria were: (a) a newly diagnosed oral cavity squamous cell carcinoma, pathologically confirmed by a biopsy; (b) an age of >18 years; and (c) signed informed consent.

Patients were not eligible if they had: (a) any contraindication to the MRI exam; (b) previous surgery or (chemo)radiotherapy treatment; and (c) the recurrence and (d) the presence of severe artifacts in both DKI and DCE-MRI.

The present study was authorized by the Institutional Review Board of our institute (approval number RS 1175/19).

### 2.2. MRI Protocol

The MRI exam was performed using a 3-T superconductive system (Discovery MR750w; GE Medical System, Waukesha, WI, USA) with a 24-channel phased array HN coil. The acquisition protocol included fast spin echo (FSE) T1-weighted images in the axial plane and T2-weighted images in the axial and coronal plane for DKI and DCE-MRI.

DKI was performed using a spin echo imaging sequence with the following parameters: acquisition matrix, 96 × 128; field of view, 24 cm × 24 cm; TR/TE, 6500/86.3 ms; slice thickness, 4.4 mm; intersection gap, 0.4 mm; and bandwidth, 1953 Hz/pixel. Five different b-values in the range of 0–2000 s/mm^2^ were used: 0, 500, 1000, 1500 and 2000 s/mm^2^. A gradient sensitization in three orthogonal directions was employed and the trace-weighted images were used to derive the DKI analysis. A different number of signal averages was chosen to compensate for the reduction in the signal-to-noise ratio (SNR) at increasing b-values: four for b-values in the interval 0–1000 s/mm^2^ and eight for b-values of 1500 s/mm^2^ and 2000 s/mm^2^.

DCE-MRI was obtained using a 3D fast spoiled gradient echo sequence with the following parameters: acquisition matrix, 162 × 150; field of view, 25 × 25 cm; TR/TE, 4.7/1.07 ms; flip angle, 30°; slice thickness, 4.4 mm; and bandwidth, 390 Hz/pixel. During the passage of the contrast agent, 70 multiple volumes were acquired with a temporal resolution of 5 s and a total scan duration of 5 min 50 s. After three volumes, the contrast agent (0.1 mmol/kg body weight of gadopentetate dimeglumine) was intravenously injected at a rate of 3 mL/s.

### 2.3. DCE-MRI and DKI Quantification

The perfusion maps were generated from DCE-MRI using commercial software (GenIQ General, GE Advanced Workstation, Palo Alto, CA). A two-compartment pharmacokinetic model [10] and an automatic population-based selection of the arterial input function were employed to derive three hemodynamic parameters: K^trans^ (the transfer constant between the plasma and the extravascular extracellular space, EES), K_ep_ (the transfer constant between the EES and plasma) and v_e_ (the fractional EES volume).

The calculation of the D and K parameters was performed at the voxel level by fitting the signal intensity variation with increasing b-values through dedicated scripts created in MATLAB (Release 2020b, MathWorks Inc., Natick, MA, USA). The polynomial model used to analyze the DKI is described by the following equation [7]:(1)$$S\left(b\right)=S_{0}\cdot e^{-b\cdot D+b^{2}\cdot D^{2}\cdot \frac{K}{6}}$$
where S(b) and S_0_ indicate the signal intensities for a diffusion weighting of b and 0 s/mm^2^, respectively. D is the diffusion coefficient (in mm^2^/s) corrected for the non-Gaussian diffusive motion and K is the diffusional kurtosis (a dimensionless parameter), which takes values larger than 0 in the presence of non-Gaussian water behavior. A non-linear constrained minimization algorithm [17] was applied to compute D and K from Equation (1).

Lower and upper bounds were used to constrain the possible solutions for D and K in order to find only clinically meaningful results: D was constrained in the range between 0.3 × 10^−3^ and 3 × 10^−3^ mm^2^/s and K values were between 0 and 3. To reduce the risk of a convergence to suboptimal solutions, multiple optimizations were executed for each fit within the voxel, with 10 randomly chosen starting values of D and K between the fixed bounds. D and K solutions outside the fixed bounds were defined as “not-a-number” and were removed from the analyses.

### 2.4. Lesion Delineation

For each patient, the segmentation of the volume of interest (VOI) was performed using the free and open-source 3D Slicer software (version 4.11) [18]. The lesion was manually delineated on a K^trans^ parametric map using the T2-weighted image as a guide for the tumor identification, excluding large vessels and bones from the VOI. The lesion contour was then transferred from the DCE-MRI maps to DKI by a rigid propagation. After a visual inspection, if required, the lesion contour was adjusted to cater for possible geometrical distortions and susceptibility artifacts, which typically affect diffusion-weighted images.

### 2.5. Pathological Analysis

A histological analysis was performed on the resection specimens, which were formalin-fixed and paraffin-embedded. Slides from all the available blocks of each case were evaluated by A.A., a pathologist in training with 3 years of experience, and R.C., a pathologist with more than 25 years of experience. A single slide per case was chosen as the most representative of the tumor. The immunohistochemistry (IHC) for CD3, CD20, CD4 and CD8 was performed by V.B. and E.G. The samples were stained with the following: a CD3 antibody at a dilution of 1:200 (clone LN10, Leica Biosystems, Wetzlar, Germany) in a pH6 citrate buffer; a CD20 antibody at a dilution of 1:50 (clone L26, Leica) in a pH6 citrate buffer; a CD4 antibody at a dilution of 1:100 (clone 4B12, Leica) in a Tris/EDTA-based buffer; and a CD8 antibody at a dilution of 1:20 (clone 4B11, Leica) in a Tris/EDTA-based buffer. The immunoreactions were revealed by a biotin-free Bond Polymer Refine Detection system (Leica Biosystem) in a Bond III autostainer (Leica Biosystems, Wetzlar, Germany). Diaminobenzidine (DAB) was used as a chromogenic substrate.

All slides were then independently evaluated by A.A. and R.C. for a series of pathological features concerning the tumor cells and TME, with a joint session to solve any disagreements. Such features included a grading according to the WHO criteria, the presence and extent of keratinization, necrosis and the inflammatory response, the pattern of growth (expansive, mixed or infiltrative), the intensity of the desmoplastic reaction, the presence of perineural and lymphovascular infiltration and the presence of peritumoral tertiary follicles. The cases were also evaluated for tumor-infiltrating lymphocytes (TILs) and for the tumor–stroma ratio (TSR), which was calculated as a percentage of the tumor area occupied by the neoplastic cells. Lastly, the immunohistochemical stains for CD3, CD20, CD4 and CD8 were reviewed; each case was defined as having either a CD3 or a CD20 prominence and either a CD4 or a CD8 prominence or balanced for these lymphocytic populations instead.

### 2.6. Statistics Analysis

Home-made scripts developed in MATLAB (Release 2020b, MathWorks Inc., Natick, MA, USA) were used to calculate the percentiles (P) P10, P25, P50, P75 and P90 as well as the skewness, kurtosis, energy and entropy values from the voxel-based distribution of each DCE-MRI and DKI parameters within the entire lesion. The same bin size was used for each patient to extract these data. The bin size was 0.2 min^−1^, 0.5 min^−1^ and 0.025 for K^trans^, K_ep_ and v_e_, respectively; it was 0.1 × 10^−3^ mm^2^/s and 0.1 (unitless) for D and K, respectively.

Median values and interquartile ranges were used to describe the continuous variables; frequencies and percentage values were used for categorical ones.

The Shapiro–Wilk test was applied in order to evaluate the normality distribution of the data. The Kruskal–Wallis test was applied to assess the differences between the groups of patients. The relationships between the DCE-MRI/DKI parameters and the immunohistochemical features were evaluated by Spearman’s rho correlation test. The relationships between the categorical variables were evaluated using the chi-squared or Fisher’s exact tests, as appropriate.

A *p* level < 0.05 was considered to be statistically significant. All statistical analyses were performed in MATLAB.

## 3. Results

The patient and tumor characteristics are reported in Table 1.

### 3.1. DCE-MRI and DKI Quantification

The summary statistics of all the selected imaging parameters are reported in Table 2. The DKI-derived parameters were not available for two patients due to the poor image quality of the DKI.

### 3.2. Pathological Results

The median (IQR) level of the TILs was 15% (25%); the median (IQR) value of the TSR was 65% (35%). The frequency distribution of the immunohistochemical factors was: grading, well-differentiated, 3 (8.1%), moderately differentiated, 19 (51.4%) and poorly differentiated, 15 (40.5%); keratinization, non-keratinizing, 1 (2.7%), focally keratinizing, 10 (27%), diffusely keratinizing, 25 (67.6%) and missing, 1 (2.7%); tumor growth pattern, expansive, 7 (18.9%), mixed, 21 (56.8%) and infiltrative, 9 (24.3%); necrosis, absent, 25 (67.6%) and focal, 12 (32.4%); inflammatory infiltrate, mild, 8 (21.6%), moderate, 17 (45.9%) and intense, 12 (32.4%); desmoplastic reaction, mild, 12 (32.4%), moderate, 12 (32.4%), intense, 12 (32.4%) and missing, 1 (2.7%); perineural infiltration, absent, 15 (40.5%) and present, 22 (59.5%); vascular invasion, absent, 32 (86.5%) and present, 5 (13.5%); peritumoral tertiary follicles, absent, 14 (37.8%), rare, 17 (45.9%) and abundant, 6 (16.2%); prevalence of CD3, 27 (73%), prevalence of CD20, 7 (18.9%), equilibrium, 2 (5.4%) and missing, 1 (2.7%); and prevalence of CD4, 26 (70.3%), prevalence of CD8, 7 (18.9%), equilibrium, 3 (8.1%) and missing, 1 (2.7%).

Significant associations were found between the grading and the desmoplastic reaction (chi-squared = 11.4, *p* = 0.023) and between the grading and the tumor growth pattern (chi-squared = 15.4, *p* = 0.004).

### 3.3. Association between Imaging Parameters and Pathological Features

Considering that most of the variables were not normally distributed (see Appendix A for details), the Kruskal–Wallis test was used to investigate the effect of the different pathological factors on the imaging parameters because the assumption about the normality was not satisfied. A number of statistically significant differences were found in both the DCE-MRI and DKI parameters in relation to the inflammatory infiltrate, keratinization, tumor grading and desmoplastic reaction (see Appendix A for details). The most relevant associations are illustrated in Figure 1. No differences emerged in the imaging parameters based on necrosis, the peritumoral tertiary follicles, vascular invasion and the tumor growth pattern.

There were also significant relationships between the TILs and several DKI-derived parameters, as shown in Table 3 and Figure 2. No association was found between the TSR and both DCE-MRI and DKI, with the exception of the IQR and energy of K_ep_, which were directly (rho = 0.342, *p* = 0.038) and inversely (rho = −0.373, *p* = 0.023) related to the TSR, respectively.

Moderately significant differences emerged in the skewness of K based on the CD3/20 prominence, which indicated a decrease in the skewness when CD20 was predominant over CD3 (*p* = 0.048); an increase in the energy of K^trans^ was also observed when CD8 was predominant over CD4 (*p* = 0.019).

Two exemplificative cases showing the relationships between the TILs and DKI-derived parameters are reported in Figure 3 and Figure 4 and two illustrative cases showing the relationships between the tumor grading and DCE-MRI are reported in Figure 5 and Figure 6.

## 4. Discussion

In this prospective study, we hypothesized that the imaging features of OSCCs, in particular those derived from perfusion- and diffusion-weighted techniques, could reflect not only the characteristics of cancer, but also the microenvironment composition and complexity, giving an insight into the association between them, as argued by previous reports [6,14,15,16]. There is a growing need for personalized risk assessments and a more comprehensive evaluation of the patient response to radio-chemotherapy and immunotherapy by non-invasive and quantitative imaging methods. In the era of precision medicine, imaging-based parameters, together with pathological factors, genomic data and clinical variables, may be used to build multifactorial prognostic models and improve the prediction of the clinical outcome in head and neck cancer patients, as suggested by the current literature [11,12,19,20].

It has already been documented that besides cell density, nuclear and stroma areas may also significantly impact on the ADC in laryngeal and hypopharyngeal carcinomas [6], suggesting that the higher pre-treatment ADC values of poorly responding patients may be attributed not only to a larger necrotic component, but also to a larger stromal area [21]. It has also been shown that both the tumor proliferation and CD3-positive cell count have a significant negative correlation with the ADC in oropharyngeal SCCs [15]; this demonstrates that the characteristics of tumor immune infiltrates may have a relevant role in determining the biological properties of the tumor microstructure.

Different from previous investigations, our correlation study included both DCE-MRI and DKI. To take into account the possible heterogeneity within a lesion both in terms of vascularity and diffusivity, we proposed a histogram-based approach to DCE-MRI and DKI maps in order to enhance the ability to find associations between the imaging-based and pathological factors [22].

We found that a number of pathological factors affected the imaging parameters, i.e., inflammatory infiltrate, keratinization, tumor grading and desmoplastic reaction. Specifically, the mean D value monotonically decreased with an increased degree of inflammatory infiltrate. This may be explained by considering that an intense inflammatory infiltrate may reduce the water molecule mobility in tissues due to the high levels of macrophages and T cells along with a consequence-associated increased viscosity [6]. More interestingly, evident relationships emerged between the DKI parameters and TILs; lower D and higher K values were associated with increased TILs. These results suggested that higher levels of tumor-infiltrating lymphocytes could cause a restriction in the water molecule mobility. This was consistent with the findings of Swartz et al. [15], who evaluated the association between the ADC and TILs in oropharyngeal carcinomas. Moreover, higher K values may be indicative of an increased tortuosity and irregularity of the water microscopic motion caused by a more complex tissue architecture, attributable to a higher density of lymphocytes. This supports the hypothesis that DKI could provide additional insights into microanatomical tumor characteristics, thanks to the quantification of non-Gaussian diffusion effects observed at high b-values (>2000 s/mm^2^) [7]. Even though DKI has been proven to be a promising technique in various oncologic applications, i.e., cancer detection, the differentiation between malignant and benign lesions or the early prediction of treatment responses [8,23], it has not yet been widely used in clinical settings, mainly because it requires an adequate SNR at high b-values, standardized acquisition protocols and more sophisticated models to fit the DW signal intensity compared with conventional DWI [24]. The present investigation also intended to contribute to a better comprehension of the D and K parameters, exploring the potential of DKI through a correlation study with histopathology.

Significant differences were found in several DCE-MRI parameters according to the degree of tumor grading, suggesting that higher-grade tumors are characterized by a reduced level of perfusion. A decreased perfusion in higher-grade tumors may be explained in terms of more extended hypoxic/hypoperfused subregions or in terms of a larger amount of fibrous connective tissue around the tumor cells in consideration of the significant association found between the tumor grading and the levels of desmoplastic reactions. Consistent with D’Urso et al. [25], who reported that K_ep_ was the most relevant DCE-derived parameter to predict the outcome in oropharyngeal SCCs, in the present investigation, K_ep_ also turned out to be the most promising hemodynamic parameter in comparison with the other DCE-derived parameters. This may be due to the fact that it is derived from the ratio between K^trans^ and v_e_ [10]; consequently, it is able to reflect not only the blood flow, blood volume and the vascular permeability, but also the characteristics of the extravascular and extracellular compartments [26], enhancing the possible associations with the TME.

Contrary to Yamada et al. [27], who found that the histologic grade in oral carcinomas had a significant inverse association with D and a significant positive association with K, we could not evidence any correlation between the DKI parameters and the tumor grading. This could be due to the differences in the characteristics of the patient population (for example, our patients were distributed differently within the groups with various histologic grades compared to Yamada et al.), differences in the sample size or discrepancies in the MRI acquisition protocols and in the image analysis methods.

No dissimilarity in the DKI parameters was found in relation to the tumor grading; this was similar to Driessen et al. [6], who did not report any differences in the ADC between moderately and poorly differentiated tumors in laryngeal and hypopharyngeal carcinomas. Even though the ADC values were not indicated in the present investigation, a very strong association has already been demonstrated between conventional ADCs and D derived from DKI in HN tumors [9,28], suggesting that they may be considered to be substantially equivalent.

Interestingly, a lower median K_ep_ value was observed in the tumors with a more intense desmoplastic reaction, indicating that a larger amount of dense fibrous connective tissue around the tumor cells was associated with reduced perfusion levels of the lesion. Differences in DKI also emerged according to the desmoplastic reaction. Tumors with higher levels of desmoplastic reactions were characterized by larger values of kurtosis of K; thus, by more homogeneous K values. However, this was not easily explained and deserves a further investigation with a larger patient population along with other associations we found that could not be straightforwardly interpreted. Similarly, we were not able to explain the moderate associations found between the CD3/20 prominence and the skewness of K and between the CD4/8 prominence and the energy of K^trans^, which need further confirmation.

We did not observe any differences in the imaging parameters based on necrosis, the peritumoral tertiary follicles, vascular invasion and the tumor growth pattern. The lack of an association between necrosis and the DCE-MRI parameters could be attributed to the small sample size and to the fact that the majority of the analyzed lesions had no necrosis (67.6% absent versus 32.4% focal). Unfortunately, in relation to these pathological features, a comparison with the literature data was not possible because no similar reports had been published.

Contrary to an earlier study of Meyer et al. [16] on HNSCCs who described several associations between v_e_ and K^trans^ and both TILs and the TSR, we did not find any significant relationships. As mentioned before, discrepancies with previous studies may have been caused by differences in the characteristics of the patient population, in the MRI protocols or in the approach to the quantitative image analysis.

This prospective study had a few limitations. The analyzed patient population was small and our findings should be confirmed with a larger dataset. The small sample size did not allow us to explore the potential of second-order MRI-based radiomics [29] to better identify the associations between the DCE-MRI/DKI texture features and specific microenvironmental factors, as recently proposed by Katsoulakis et al. based on CT images [30]. The MRI analysis was derived from the entire tumor volume whereas the histopathologic results were derived from the most representative slide of the tumor. This could have introduced a bias due to the lack of an accurate spatial registration between the histological section and the MRI volumetric analysis [6]. Moreover, the analysis of pathological features such as the intensity of the inflammatory infiltrate, the degree of the desmoplastic reaction and even the quantification of TILs and the definition of the tumor grade is bound to be prone to subjective errors and low inter-observer concordance. Lastly, in our patient cohort, the different tumor subsites were not equally populated; tongue cancer was the most represented. However, this distribution reflected the current characteristics and epidemiology of OSCCs in the general population, where tongue cancer is the most common of all oral carcinomas [2].

## 5. Conclusions

Non-invasive MRI biomarkers from DCE-MRI and DKI were significantly associated with a number of pathological factors such as the inflammatory infiltrate, tumor grading, desmoplastic reaction and TILs. Although further investigations are needed on a larger dataset, these findings contribute to a more comprehensive biological characterization of OSCCs and could be used in future investigations for diagnostic and prognostic purposes.

## Figures and Tables

**Figure 1 cancers-15-00015-f001:**
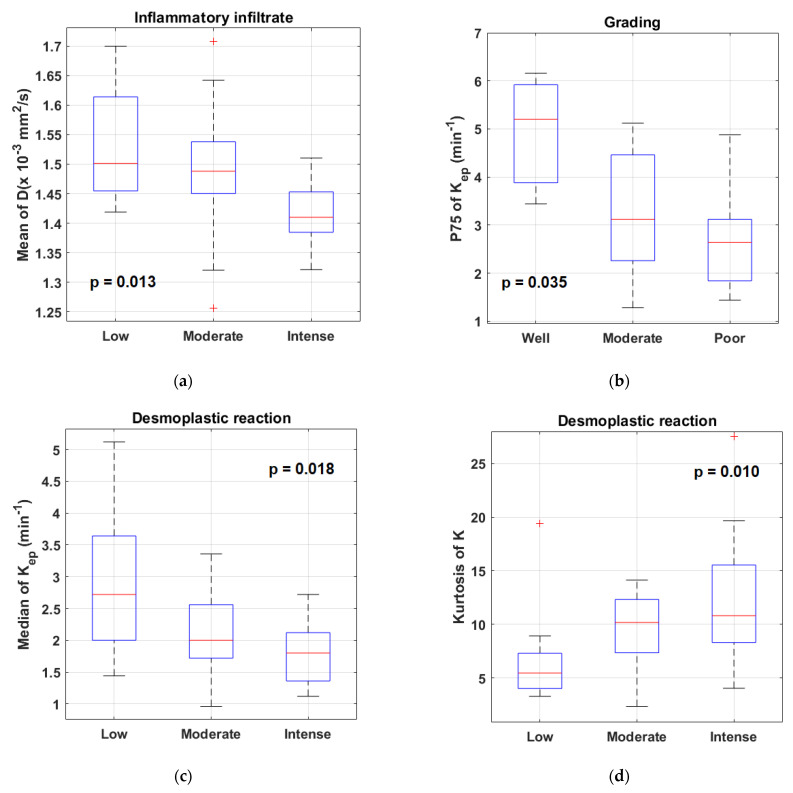
Comparison of mean D values between different groups according to the degree of inflammatory infiltrate (**a**); comparison of P75 values of K_ep_ between different groups according to the degree of tumor grading (**b**); comparison of median K_ep_ values and kurtosis values of K between different groups according to the level of desmoplastic reaction ((**c**) and (**d**), respectively). Outliers are plotted with square markers.

**Figure 2 cancers-15-00015-f002:**
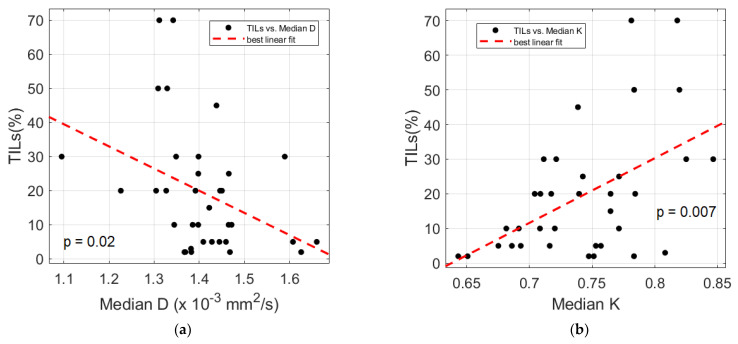
Scatter plots of percentage TILs versus median D (**a**) and median K (unitless) (**b**). The *p*-values for the Spearman correlation tests are indicated for each plot and the best linear polynomial fit is depicted (dashed red line).

**Figure 3 cancers-15-00015-f003:**
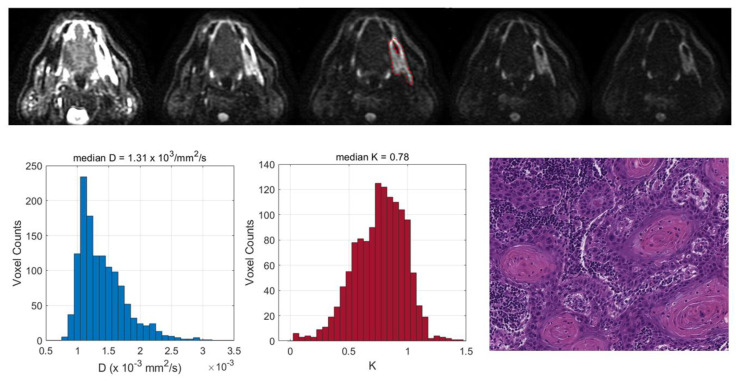
Top (from left to right): central section of a retromolar trigone tumor in a 62-year-old woman on diffusion-weighted images obtained with increasing values: *b* = 0, 500, 1000, 1500 and 2000 mm^2^/s; user-defined lesion contour is in red. Bottom (from left to right): histograms of D and K values derived from the entire tumor volume, indicating both a restriction in the water molecule diffusivity and increased kurtosis as well as the presence of high levels of tumor-infiltrating lymphocytes (TILs) (hematoxylin–eosin; original magnification: 20×).

**Figure 4 cancers-15-00015-f004:**
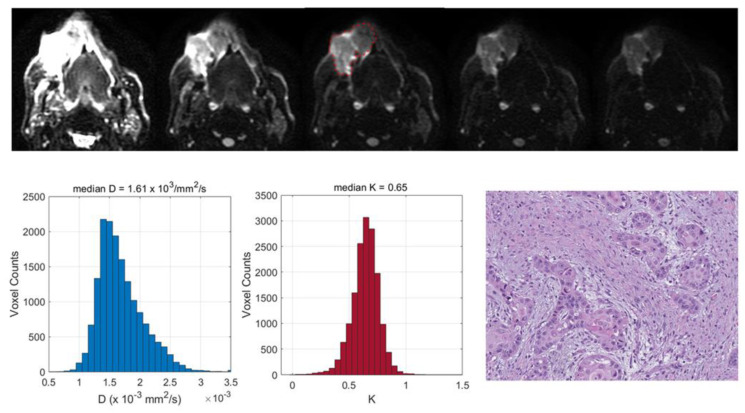
Top (from left to right): central section of a cheek tumor in a 68-year-old man on diffusion-weighted images obtained with increasing values: *b* = 0, 500, 1000, 1500 and 2000 mm^2^/s; the user-defined lesion contour is in red. Bottom (from left to right): histograms of D and K values derived from the entire tumor volume, indicating slightly restricted water molecule diffusivity and moderately increased kurtosis as well as the presence of low levels of tumor-infiltrating lymphocytes (TILs) (hematoxylin–eosin; original magnification: 20×).

**Figure 5 cancers-15-00015-f005:**
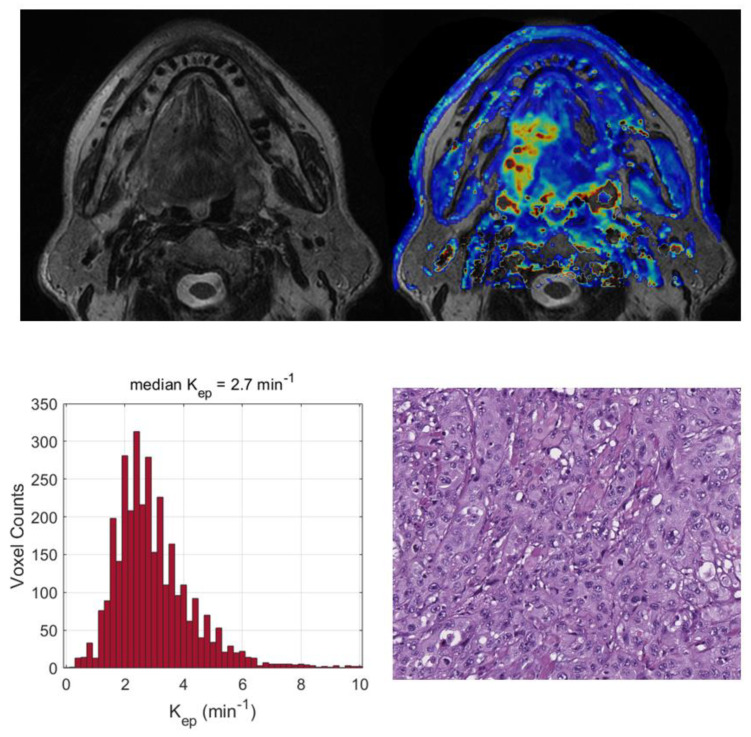
Top (from left to right): central section of a tongue lesion in a 69-year-old man on T2-weighted images (a) and on the corresponding map of K_ep_ (min^−1^), showing a moderately vascularized tumor (median K_ep_ value = 2.7 min^−1^). Bottom (from left to right): histograms of K_ep_ values derived from the entire tumor volume and poorly differentiated squamous cell carcinoma (hematoxylin–eosin; original magnification: 20×).

**Figure 6 cancers-15-00015-f006:**
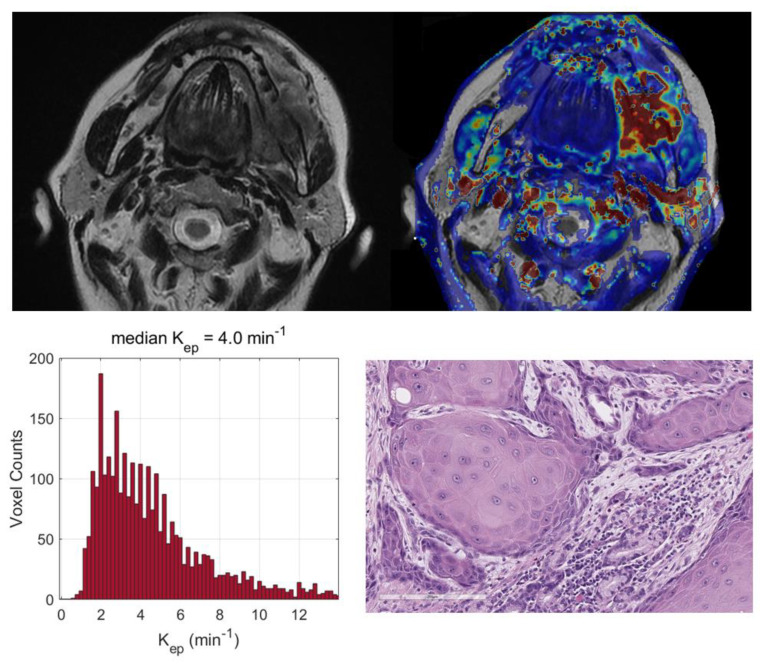
Top (from left to right): central section of a retromolar trigon lesion in a 63-year-old woman on T2-weighted images and on the corresponding map of K_ep_ (min^−1^), showing a well-vascularized tumor (median K_ep_ value = 4.0 min^−1^). Bottom (from left to right): histograms of K_ep_ values derived from the entire tumor volume and a well-differentiated squamous cell carcinoma (hematoxylin–eosin; original magnification: 20×).

**Table 1 cancers-15-00015-t001:** Selected patient and tumor characteristics.

Characteristic	Parameter
Patient number	37
Sex (M/F)	23 (62.2%)/14 (37.8%)
Age (years)	
Mean (range)	64 (34–91)
Primary tumor subsite	
Tongue	24 (64.9%)
Oral floor	3 (8.1%)
Retromolar trigone	3 (8.1%)
Cheek	4 (10.8%)
Hard palate	3 (8.1%)
T stage	
T1	2 (5.4%)
T2	4 (10.8%)
T3	19 (51.4%)
T4 (a, b)	12 (32.4%)
N stage	
N0	13 (35.1%)
N1	10 (27.0%)
N2 (b, c)	6 (16.2%)
N3b	8 (21.6%)
Volume (cm^3^)	
Mean (range)	17.3 (1–87)

**Table 2 cancers-15-00015-t002:** Summary statistics of the DCE-MRI and DKI-derived parameters.

**K^trans^**	**Median**	**IQR**	**P10**	**P25**	**P75**	**P90**	**Skewness**	**Kurtosis**	**Mean**	**Std**	**Energy**	**Entropy**
Median	1.46	0.90	0.69	0.98	1.98	2.60	1.29	5.24	1.63	0.83	0.10	3.74
IQR	0.91	0.78	0.46	0.60	1.39	1.98	1.16	5.79	0.95	0.57	0.08	1.01
**K_ep_**	**Median**	**IQR**	**P10**	**P25**	**P75**	**P90**	**Skewness**	**Kurtosis**	**Mean**	**Std**	**Energy**	**Entropy**
Median	2.08	1.52	1.04	1.36	3.12	4.56	7.28	70.93	2.49	2.44	0.13	3.34
IQR	1.14	1.16	0.46	0.62	2.12	2.38	5.11	114.57	1.76	3.30	0.09	0.91
**v_e_**	**Median**	**IQR**	**P10**	**P25**	**P75**	**P90**	**Skewness**	**Kurtosis**	**Mean**	**Std**	**Energy**	**Entropy**
Median	0.69	0.37	0.34	0.48	0.92	1.00	−0.25	2.39	0.67	0.24	0.06	4.73
IQR	0.24	0.19	0.19	0.18	0.21	0.03	0.76	1.55	0.18	0.07	0.06	0.72
**D**	**Median**	**IQR**	**P10**	**P25**	**P75**	**P90**	**Skewness**	**Kurtosis**	**Mean**	**Std**	**Energy**	**Entropy**
Median	1.40	0.53	1.00	1.14	1.68	2.02	0.96	4.43	1.46	0.42	0.07	4.00
IQR	0.11	0.08	0.09	0.10	0.15	0.18	0.49	1.39	0.10	0.06	0.01	0.21
**K**	**Median**	**IQR**	**P10**	**P25**	**P75**	**P90**	**Skewness**	**Kurtosis**	**Mean**	**Std**	**Energy**	**Entropy**
Median	0.75	0.27	0.48	0.61	0.88	0.99	0.59	8.54	0.74	0.23	0.14	3.17
IQR	0.07	0.07	0.05	0.04	0.07	0.09	0.72	5.29	0.07	0.04	0.03	0.36

P10/P25: 10th/25th percentiles, respectively; Std: standard deviation; K^trans^: transfer constant between plasma and extravascular extracellular space (EES) (min^−1^); K_ep_: transfer constant between EES and plasma (min^−1^); v_e_: fractional volume of EES (fractional units); D: diffusion coefficient (10^−3^ mm^2^/s); K: diffusional kurtosis parameter (dimensionless).

**Table 3 cancers-15-00015-t003:** Correlation coefficients between the DKI-derived parameters and TILs.

**D**	**Median**	**IQR**	**P10**	**P25**	**P75**	**P90**	**Skewness**	**Kurtosis**	**Mean**	**Std**	**Energy**	**Entropy**
*Rho*	−0.393	−0.073	−0.295	−0.447	−0.429	−0.399	0.012	0.088	−0.521	−0.175	0.138	−0.174
*p-Value*	**0.02**	0.678	**0.086**	**0.007**	**0.01**	**0.018**	0.947	0.617	**0.001**	0.316	0.43	0.316
**K**	**Median**	**IQR**	**P10**	**P25**	**P75**	**P90**	**Skewness**	**Kurtosis**	**Mean**	**Std**	**Energy**	**Entropy**
*Rho*	0.445	0.328	−0.225	0.307	0.444	0.38	−0.213	−0.18	0.372	0.336	−0.371	0.364
*p-Value*	**0.007**	0.054	0.194	0.073	**0.008**	**0.024**	0.218	0.3	**0.028**	**0.048**	**0.028**	**0.031**

TILs: tumor-infiltrating lymphocytes; P10/P25: 10th/25th percentiles, respectively; Std: standard deviation; D: diffusion coefficient; K: diffusional kurtosis parameter. Statistically significant *p*-values are in bold.

## Data Availability

The original contributions presented in the study are included in the article/Appendix A. Further inquiries can be directed to the corresponding author.

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
