# Peer review of "Microenvironmental Factors in Oral Cavity Squamous Cell Carcinoma Undergoing Surgery: Correlation with Diffusion Kurtosis Imaging and Dynamic Contrast-Enhanced MRI"

_cancers, 2022, doi:10.3390/cancers15010015_

Round 1
Reviewer 1 Report
A prospective study on a small number of paptients which is certainly interesting for a limited readership:
1: Usually, oral (cavity) squamos cell carcinoma ist abbreviated with OSCC. This term is generally used in literature.
2: The authors state that "Magnetic resonance imaging (MRI) is the modality of choice for staging, follow-up and response treatment evaluation": This is usually not the case, especially for tumors with infiltration of the jaw, a CT is needed. This part needs to be corrected. It rather is one possible technique of imaging.
3. I would remove "T0" out of the charcteristics. Here, no invasive tumor is diagnosed.
4. Figure 1 does not show p-values.
5. The potential clinical benefict should be marked out.
It is an interesting topic, however the readership is limited, this paper would probably fit rather in a more specialized journal.
Author Response
The authors thank the editor and the reviewers for their constructive criticism and suggestions.
In response to Reviewer 1:
1: Usually, oral (cavity) squamous cell carcinoma is abbreviated with OSCC. This term is generally used in literature.
R: We have now used the abbreviation OSCC instead of OCSCC, as suggested.
2: The authors state that "Magnetic resonance imaging (MRI) is the modality of choice for staging, follow-up and response treatment evaluation": This is usually not the case, especially for tumors with infiltration of the jaw, a CT is needed. This part needs to be corrected. It rather is one possible technique of imaging.
R: As suggested, we have now modified this sentence in the Introduction (Page 2, lines 6-10) and we have added a new reference [Mahajan A.; Ahuja A.; Sable N, Stambuk HE. Imaging in oral cancers: A comprehensive review. Oral Oncology 2020, 104:104658 https://doi.org/10.1016/j.oraloncology.2020.104658].
- I would remove "T0" out of the characteristics. Here, no invasive tumor is diagnosed.
R: We have now removed “T0” from the characteristics of Table 1, as suggested.
- Figure 1 does not show p-values.
R: We thank the Reviewer 1 for this comment. We have now reported the p-values in each box-plot of Figure 1.
- The potential clinical benefits should be marked out.
R: We thank the Reviewer 1 for this comment. We are aware that the present study is preliminary and further investigation is needed in order to investigate the potential clinical benefits of our findings. The clinical utility of advanced MRI techniques and the prognostic value of tumor microenvironment characteristics had already been mentioned in Introduction, specifying that unfortunately only a few investigations have been conducted to assess the associations between imaging-based parameters and pathological factors, although both have emerged as useful biomarkers in head and neck cancer. We agree that the potential clinical benefits of the present investigation could be more clearly marked out, thus we have added a new paragraph in Discussion (Page 11, Lines 5-12 of Discussion), and two new references, to better address this point [ref19: Wu, M.;, Zhang, Y.; Zhang, Y.; Liu, Y.; Wu, M.; Ye, Z. Imaging-based Biomarkers for Predicting and Evaluating Cancer Immunotherapy Response. Radiol Imaging Cancer. 2019, 1:e190031. https://doi.org/: 10.1148/rycan.2019190031; ref20: Borsetto, D.; Tomasoni, M.; Payne, K. et al. Prognostic Significance of CD4+ and CD8+ Tumor-Infiltrating Lymphocytes in Head and Neck Squamous Cell Carcinoma: A Meta-Analysis. Cancers (Basel). 2021, 13:781. https://doi.org/: 10.3390/cancers13040781].
- It is an interesting topic, however the readership is limited, this paper would probably fit rather in a more specialized journal
R: While advanced MRI acquisition protocols and image analyses have been used, we believe that the present study could be of interest to the readers of this journal, as it was conducted involving a multidisciplinary group of professionals and it may also be helpful for research on similar topics in different tumor sites.

Reviewer 2 Report
Dear authors, I have carefully read the article on how MRI imaging parameters can correlate with the composition of the oral cavity carcinoma microenvironment. This could give a new prognostic role to magnetic resonance, already used in the preoperative evaluation of this type of tumor, without requiring other resources.
The number of patients is unfortunately very low and more than half of the cases are represented by tongue carcinoma. However the results are interesting. The purely muscular composition of the tongue, different from the other subsites, could give different results compared to the study of the other subsites, although examples of carcinoma originating from other sites are reported in the article.
Author Response
The authors thank the editor and the reviewers for their constructive criticism and suggestions.
In response to Reviewer 2:
Dear authors, I have carefully read the article on how MRI imaging parameters can correlate with the composition of the oral cavity carcinoma microenvironment. This could give a new prognostic role to magnetic resonance, already used in the preoperative evaluation of this type of tumor, without requiring other resources.
The number of patients is unfortunately very low and more than half of the cases are represented by tongue carcinoma. However the results are interesting. The purely muscular composition of the tongue, different from the other subsites, could give different results compared to the study of the other subsites, although examples of carcinoma originating from other sites are reported in the article.
R: We thank the Reviewer #2 for this comment. We are aware that a larger dataset would be needed to corroborate our findings and further investigate the generalization of our results, as already mentioned in the limitations of the study. We enrolled all OSCC patients referring to our institute from 2018 to 202, but due to both the relative rarity of this kind of tumor and the inclusion/exclusion criteria, a homogeneous cohort of only 37 patients was identified. We are aware of this limitation and we have now reported the need for further analyses on a larger population also in “Conclusions”.
With regard to the tongue subsite, we acknowledge that the oral cavity has a degree of heterogeneity when it comes to the tissue composition of its subsites, but we feel like our cohort reflects the characteristics and epidemiology of OSCC in the general population, and represents a noticeably homogenous group per se, focused on the oral cavity alone rather than the whole head and neck district.
From an anatomopathological standpoint, according to the guidelines reported in the WHO classification of head and neck tumors (WHO Classification of Tumours Editorial Board. Head and neck tumours (WHO classification of tumours series, 5th ed. vol. 9). Lyon (France): International Agency for Research on Cancer 2022), squamous cells carcinomas originating in the oral cavity are grouped together as OSCCs, with the different subsites being cited for their differences only when talking about epidemiology and stage at diagnosis - and therefore mortality - with tongue carcinomas often presenting with positive lymph nodes. On the other hand, we found that according to both literature and our experience, no relevant differences can be identified among subsites in terms of histological variables such as TILs, grading, desmoplastic reaction, etc. However, once the number of our patients will have increased, it may certainly be of interest to evaluate the data taking the single subsites into consideration. In the light of this comment, we have now mentioned the uneven distribution of tumor subsites in our patient population as a further limitation of the study.

Round 2
Reviewer 1 Report
The authors now added relevant information and addressed initial concerns.
I would suggest to accept the manuscript in the present form.